# Self-triggered Distributed Formation Control of Under-actuated Unmanned Surface Vehicles in GPS-denied Environments

Xintong He
School of
Marine Electrical Engineering
Dalian Maritime University
Dalian, China
1471252118@qq.com

Lu Liu*
School of
Marine Electrical Engineering
Dalian Maritime University
Dalian Maritime University
Dalian, China
luliu@dlmu.edu.cn

Haoliang Wang
School of
Marine Electrical Engineering
Dalian Maritime University
Dalian, China
haoliang.wang12@dlmu.edu.cn

Zhouhua Peng
School of
Marine Electrical Engineering
Dalian Maritime University
Dalian, China
zhpeng@dlmu.edu.cn

Dan Wang
School of
Marine Electrical Engineering
Dalian Maritime University
Dalian, China
dwangdl@gmail.com

*Abstract*—This paper presents a self-triggered distributed formation control method for multiple unmanned surface vehicles (USVs) with limited communication resources in GPS-denied environments. Initially, we configure a velocity quantization-based self-triggered mechanism to reduce the data exchange rate through the wireless network infrastructure. As information transmitted between vehicles is indexed rather than raw data, this approach enhances security and makes data transmission more efficient and stable. Subsequently, we develop a self-triggered distributed guidance law that functions independently of the position information of one's own vehicle as well as that of neighboring vessels. Finally, simulation results validate the effectiveness of the proposed method and highlight its potential applications in environments with weak or nonexistent GPS signals.

*Index Terms*—Self-triggered communication, Distributed formation control, GPS-denied, Quantized velocity measurements, Under-actuated Unmanned surface vehicles (USVs).

## I. Introduction

In recent years, unmanned surface vehicles (USVs) have witnessed a notable surge in various fields from industry to military in recent years [1]–[4]. The distributed formation control of USVs has been getting a lot of attention owing to its recognized potential to significantly enhance operational efficiency when compared to the performance of individual USVs [5]–[8]. The fundamental concept of distributed formation control involves multiple vehicles to maintain a predesigned geometric formation while following a predetermined target.

Over the past two decades, an array of strategies for formation control has been developed, encompass-

ing approaches like the leader-follower technique, virtual structural methods, and behavioral-based strategies [9]–[11]. Among these strategies, the leader-follower technique for managing formations has seen extensive application in USVs, noted for its straightforwardness and expansibility, as mentioned in references [11]–[13]. Reference [11] introduces a sophisticated scheme for leader-follower formation control that is predicated on the positional information of the leading unit and a prearranged formation pattern. The work in [12] details the introduction of an observer-driven control technique designed to establish a containment formation swiftly. Concurrently, [13] tackles the combined issue of steady formation and trajectory tracking with a streamlined, adaptive control structure. In the literature above, actual velocity information needs to be transmitted, and quantization problems are not mentioned. In [14], to save the restricted communication bandwidth, hysteretic quantization has been utilized. However, its controller needs continuous information from neighbors.

To alleviate the communication load on vehicular wireless networks, some event-triggered and self-triggered control methods have been proposed to reduce data transmission frequency. In [15], for the problem of path following for USVs, a dynamic controller is further improved by considering the event-triggered condition. There have also been some achievements in applying event-triggered control to distributed control for multiple USVs [16], [17].

In conventional operating systems, event-triggered mechanisms are typically adequate; however, under certain circumstances, such as in large-scale or highly dynamic environments, their utilization may lead to resource inef-

Identify applicable funding agency here. If none, delete this.

ficiencies. In contrast, self-triggering represents an active communication paradigm where the next triggering instance can be forecasted based on data from the preceding event. Due to its reliance solely on the local information state of individual agents to ascertain the subsequent triggering event, self-triggering finds favorable applications in distributed control scenarios. In [18], a self-triggered cooperative vector field is developed to avoid continuous listening. In [19], a self-triggered path updating law is designed to reduce the sampling times. The self-triggering mechanisms in [18] and [19] need to estimate the states of neighbors, which increases the computational burden on vehicles. Since one's own and neighbors' position information are all needed to compute the next triggering time, these mechanisms cannot be applied in GPS-denied environments.

Inspired by the preceding discussions, this article is motivated by three key factors. Firstly, although event-triggering is extensively utilized in distributed formation control for multiple USVs, self-triggering is a more suitable approach for such systems and practical operational environments, yet it remains relatively unexplored. Secondly, the majority of self-triggering mechanisms require the reception of neighboring state information at sampling intervals, which may hinder their effectiveness in environments with weak GPS signals. Therefore, it is imperative to devise a novel triggering strategy capable of predicting the next triggering time without relying on absolute states. Thirdly, given the practical limitations of communication bandwidth, quantization cannot be overlooked. Additionally, transmitting the index of quantized information, rather than actual data, to each neighbor significantly mitigates the risk of data leakage.

The contributions of this method can be summarized as follows:

1) In contrast to the event-triggered distributed methods presented in [16], [17], a self-triggered distributed guidance law is developed, eliminating the need for continuous listening. Each vehicle calculates its next triggering time and communicates it to neighboring USVs during the current triggering event. To the best of our knowledge, this is the first attempt to consider the self-triggering approach in the design of distributed formation controllers for USVs.

2) In contrast to previous self-triggering mechanisms proposed for the formation of USVs, such as those in [18], [19], we propose a self-triggering method that focuses on relative states, rather than absolute states. Each vehicle is equipped with distance sensors to measure the relative states between its own ship and its neighbors. Therefore, we do not need to estimate the states of all neighbors, which reduces the computational burden on the vehicles. Combined with the above design that eliminates the need to transmit absolute states, the method proposed in this paper can be applied to challenging environments.

3) In contrast to the triggering methods described in [15]–[19], each vehicle needs to transmit the index of its own quantized velocity to neighbors and decode the indices received from others. The information transmitted between vessels is the index, not the actual data, which can reduce the risk of data leakage. Furthermore, since the information is already quantized at the sensing stage, it can be transmitted without any errors, even through channels with finite capacity.

The structure of this paper is organized as following. In Section II, key background concepts are introduced, and the addressed problem is formally defined. The proposed strategy is then presented in Section III, with details provided on the design of a novel self-triggered mechanism and algorithm. Simulations discussed in Section IV aid in evaluating the method's performance, accompanied by results and analysis. Lastly, Section V draws conclusions for this paper.

## II. Preliminaries and problem formulation

### A. Notations

In this document, we employ the following symbols: $\mathbb{R}^N$ stands for the n-dimensional Euclidean space; the term $\text{diag}(\cdot)$ is used to indicate a diagonal matrix. Each vehicle is assigned a label $i \in \mathcal{N} := \{1, 2, ..., N\}$. We denote by $\mathcal{N}_i$ the set of all neighbors of vehicle $i$. The adjacency matrix is defined as $\mathcal{A} = [a_{ij}] \in \mathbb{R}^{N \times N}$ with $a_{ij} = 1$, if $i,j$ are neighbors; and $a_{ij} = 0$, otherwise. The Laplacian matrix is defined as $\mathcal{L} = \mathcal{D} - \mathcal{A}$ with $\mathcal{D} = \text{diag}\{d_1, ..., d_N\}$. Meanwhile, we define $d_i = \sum_{j=1}^{N} a_{ij}, i = 1, ..., N$. For a symmetric matrix $\mathcal{L} \in \mathbb{R}^{N \times N}$ with eigenvalues $\lambda_1 \geq \lambda_2 \geq ... \geq \lambda_N$ arranged such that $\lambda_1 \geq \lambda_2 \geq ... \geq \lambda_N$, we define $\lambda_2(\mathcal{L}) := \lambda_2$.

The Lambert $W$- function, expressed as $W(y)$ for $y \geq 0$, specifically corresponds to the non-negative solution $x$ which satisfies the transcendental equation $xe^x = y$. In this paper, the subsequent fact would be consistently used without comment. By defining $a, c > 0$, the solution $x = x^*$ of the transcendental equation $a(x-b) = e^{-cx}$ can be written as

$$x^* = \frac{1}{c} W \left( \frac{ce^{-cb}}{a} \right) + b. \qquad (1)$$

### B. Problem formulation

Consider a network of underactuated USVs labeled from 1 to N, the motion of the $i$th USV is represented by a kinematic model

$$\begin{cases} \dot{x}_i & = u_i \cos\psi_i - v_i \sin\psi_i \\ \dot{y}_i & = u_i \sin\psi_i + v_i \cos\psi_i \\ \dot{\psi}_i & = r_i, \end{cases} \qquad (2)$$

where $\eta_i = [x_i, y_i]^T$ represents the position in an earth-fixed inertial frame; $\psi_i$ denotes the heading angle; $u_i$, $v_i$, and $r_i$ denote the velocities in the surge, sway, and yaw directions in a body-fixed frame (as illustrated in Fig. 1).

A graphical representation depicting the distributed formation control for USVs is presented in Fig. 1. For

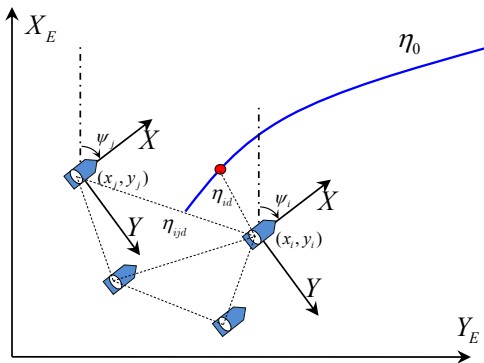

Fig. 1. Reference frames

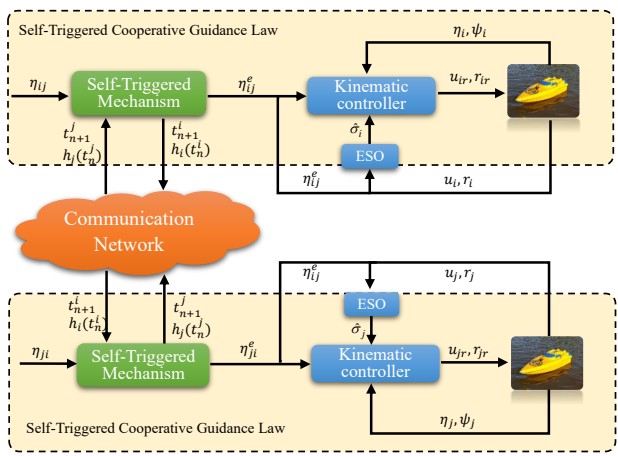

Fig. 2. The structure of the proposed distributed coordinated control based on self-triggered mechanisms.

distributed formation control, each vessel coordinates its actions solely by communicating with neighboring information, thereby achieving collaborative operation and formation arrangement of the entire system.

The control objective of the paper is

$$\lim_{t\to\infty} |(\eta_i(t) - \eta_{id}(t) - \eta_0(t)) \le l_1|, \tag{3}$$

where $\eta_i = [x_i, y_i]^T$; $\eta_0 = [x_0, y_0]^T \in \mathbb{R}^2$, the path of the leader; $\eta_{id}(t) = [x_{id}, y_{id}]^T \in \mathbb{R}^2$, a positional divergence from the leader; $l_1 \in \mathbb{R}$ is a positive constant.

To move on, the following assumptions are needed.

Assumption 1: The velocities of the USV are bounded, that is, $|u_i| \le u^*$, $|v_i| \le v^*$, and $|r_i| \le r^*$. $u^*, v^*, r^*$ are positive constants.

Assumption 2: The positional divergence $\eta_{id}$ satisfies that $\|\dot{\eta}_{id}\| \le \eta_d^*$, $\|\ddot{\eta}_{id}\| \le \eta_{dd}^*$ with $\eta_d^*$ and $\eta_{dd}^*$ being positive constants.

Given that the USV is a mechanical system, it adheres to the principles of Newton's second law. This adherence inherently limits the possible velocity and acceleration profiles of the USV. Therefore, the conditions set out in Assumption 1 are well justified. The literature, specifically Guo et al. [20], provides established constraints on velocities and accelerations that align with these principles.

### III. Design and Analysis

In this section, a joint design method combining quantization and self-triggered sampling is proposed for multiple USVs, enabling asymptotic consensus while avoiding Zeno behaviors. The architecture of the strategy presented in this study is depicted in Fig. 2. The distributed kinematic control law is designed to generate the control inputs for each USV based on self-triggered sampling of relative states and the estimated information obtained by an extended state observer.

#### A. Triggering mechanism

1) Quantization scheme: Let $P \in \mathbb{N}$ be the number of quantization levels, which is an odd number, *i.e.*, $P = 2P_0 + 1$ for some $P_0 \in \mathbb{N}_0$ and $V_0 \in \mathbb{N}$ be a

quantization range. We apply uniform quantization to the interval $[-V_0, V_0]$. More precisely, a quantization function $Q_{V_0, P}$ is defined by

$$Q_{V_0, P}[m] := \begin{cases} \frac{2sV_0}{P} & \text{if } \frac{(2s-1)V_0}{P} < m \le \frac{(2s+1)V_0}{P} \\ 0 & \text{if } -\frac{V_0}{P} \le m \le \frac{V_0}{P} \\ -Q_{V_0, P}[-m] & \text{if } m < -\frac{V_0}{P}, \end{cases} \tag{4}$$

for $-V_0 \le m \le V_0$, where $s = 1, 2, ..., P_0$.

2) Computation of inter-event times: To study the consensus of multi-USV systems, the following assumption is needed.

Assumption 3: There is a boundary $G_0 > 0$ satisfying

$$\left| \eta_i(t_0) - \frac{1}{N} \sum_{j \in \mathcal{N}} \eta_j(t_0) \right| \le [G_0, G_0]^T, \quad i \in \mathcal{N},$$

is known by all USVs, where $t_0$ is the initial time.

A range $G$ is defined by the following function

$$G(t) := 2\Gamma_\infty G_0 e^{-\omega t} + 2M/\gamma, \quad t \ge 0, \tag{5}$$

where $M \ge max\{u^*, v^*, r^*\}$, $\gamma \le \lambda_2(\mathcal{L})$; $\omega > 0$, a given decay parameter .

We denote the sampling times of vehicle $i$ by $\{t_n^i\}_{n \in \mathbb{N}}$ with $t_0^i := 0$ and $\inf_{n \in \mathbb{N}_0}(t_{n+1}^i - t_n^i) > 0$. According to [21], the following unsaturation condition (6) is satisfied at time $t = t_n^i$ for some $n \in \mathbb{N}$ under Assumption 3:

$$|\eta_{ij}(t_n^i) - \eta_{ijd}(t_n^i)| \le G(t_n^i)[1, 1]^T, \tag{6}$$

where $\eta_{ijd}(t_n^i) = \eta_{id}(t_n^i) - \eta_{jd}(t_n^i)$, $\eta_{ij}(t_n^i) = \eta_i(t_n^i) - \eta_j(t_n^i)$, for all $i \in \mathcal{N}$ and $j \in \mathcal{N}_i$. Therefore, the multi-USV system reaches a consensus at an exponential pace characterized by a decay rate.

Let $\{t_n^i\}_{n \in \mathbb{N}}$ be the sampling times of vehicle $i$ with $t_0^i := 0$. We define

$$f_i(t) := \sum_{j \in \mathcal{N}_i} (\eta_i(t) - \eta_j(t)) - \sum_{j \in \mathcal{N}_i} (\eta_i(t_n^i) - \eta_j(t_n^i)). \tag{7}$$

Using the range $G(t)$, we computate inter-event times $\tau_n^i$ of vehicle $i \in \mathcal{N}$ by

$$\begin{cases} \tau_n^i = \min\{\tau_{xn}^i, \tau_{yn}^i, \tau_{\max}^i\} \\ \begin{bmatrix} \tau_{xn}^i \\ \tau_{yn}^i \end{bmatrix} := \inf \left\{ \tau > \begin{bmatrix} \tau_{\min}^i \\ \tau_{\min}^i \end{bmatrix} : |f_n^i(\tau)| > \delta_i G(t_n^i + \tau) \right\}, \end{cases} \tag{8}$$

where $\tau_{xn}^i, \tau_{yn}^i \in \mathbb{R}$, $\delta_i > 0$ is a given threshold and $\tau_{\min}^i$, $\tau_{\max}^i > 0$ are the lower and upper bounds of inter-event times, respectively, i.e., $\tau_{\min}^i \leq \tau_n^i \leq \tau_{\max}^i$. Thus, we define the $(n+1)$th triggering time $t_{n+1}^i$ of vehicle $i \in \mathcal{N}$ by

$$t_{n+1}^i := t_n^i + \tau_n^i. \tag{9}$$

Due to the need for vehicles to communicate velocity information with each other, the quantization function $Q_{V_0, P}$ is used to quantize velocities at triggering time. We define

$$h_i(t_n^i) := Q_{V_0, P}[\dot{\eta}_i(t_n^i)], \quad i \in \mathcal{N}, \tag{10}$$

where $V_0 \geq \max\{u^* + v^*\}$. In fact, $h_i(t_n^i)$ belongs to the finite set

$$\left\{ \frac{2sV_0}{P} : s \in \mathbb{Z}, -P_0 \leq s \leq P_0 \right\}.$$

An encoder of vehicle $i$ assigns an index to each $2sV_0/P$ and transmits to a decoder of each neighbor the index corresponding to the $h_i(t_n^i)$ at triggering time. Since vehicles share $V_0$, $P$, the decoder can generate $h_i(t_n^i)$ from the received index without any errors even through finite capacity, which is then used by neighbors to compute the next triggering time.

We design the set $\{t_\ell\}_{\ell \in \mathbb{N}_0}$, which consists of sampling times for all the vehicles without duplication, while $n_i(\ell)$ denotes the count of sampling instances of vehicle $i$ within the interval $(0, t_\ell]$. Next, we describe how vehicles compute sampling times in a self-triggered fashion.

Step 1. Let $i \in \mathcal{N}$ and $n \in \mathbb{N}$. At time $t = t_n^i := t_{\ell_0}$, vehicle $i$ executes the subsequent set of actions.

i) Upon receiving an index from a neighboring vehicle at the instant $t = t_{\ell_0}$, the index is decoded, and subsequently, the velocity data of the respective neighboring vehicles is updated accordingly.

ii) Compute $\tau_{n,0}^i$ as follows. According to the triggering condition (8), we define

$$\begin{bmatrix} \widetilde{\tau}_{xn,0}^i \\ \widetilde{\tau}_{yn,0}^i \end{bmatrix} = \inf \left\{ \tau > 0 : \left| \tau d_i h_i(t_n^i) - \tau \sum_{j \in \mathcal{N}_i} h_j(t_{n_j(\ell_0)}^j) \right| \right.$$
$$\left. > \delta_i e^{-\omega \tau} G(t_{\ell_0}). \tag{11}$$

By the Lambert $W$-function $W$ (1), we define a function $\phi$ as

$$\phi(a, b, c) := \begin{cases} \frac{1}{\omega} W\left(\frac{\omega b}{|a|} e^{\frac{\omega c}{a}}\right) - \frac{c}{a} & \text{if } a \neq 0 \\ \infty & \text{if } a = 0, \end{cases} \tag{12}$$

for $a, c \in \mathbb{R}$ and $b \geq 0$.

Then (11) can be rewritten as

$$\begin{bmatrix} \widetilde{\tau}_{xn,0}^i \\ \widetilde{\tau}_{yn,0}^i \end{bmatrix} = \phi(a_{n,0}^i, b_{n,0}^i, c_{n,0}^i), \tag{13}$$

where

$$\begin{aligned} a_{n,0}^i &:= d_i h_i(t_n^i) - \sum_{j \in \mathcal{N}_i} h_j\left(t_{n_j(\ell_0)}^j\right) \\ b_{n,0}^i &:= \delta_i G(t_{\ell_0}) [1, 1]^T \\ c_{n,0}^i &:= [0, 0]^T, \end{aligned} \tag{14}$$

and we can obtain the $\tau_{n,0}^i$ by

$$\tau_{n,0}^i = \min\left\{\widetilde{\tau}_{xn,0}^i, \widetilde{\tau}_{yn,0}^i, \tau_{\max}^i\right\}. \tag{15}$$

iii) Set $k = 0$.

iiii) Send information to neighbors. The quantized velocity information $h_i(t_n^i)$ of the own vehicle is encoded into an index belonging to a finite set of cardinality $2P_0 + 1$, and this index is then communicated to each neighboring vehicle.

Step 2. Vehicle $i$ plans to activate the sensor at time $t = t_n^i + \tau_{n,k}^i$.

Step 3-a. If vehicle $i$ receives information from some neighbor in the interval $(t_{\ell_k}, t_n^i + \tau_{n,k}^i)$, $i$ needs to compute a candidate of the next triggering time again. In this situation, let $k \in \mathbb{N}$ and $i$ receive new signals from its neighbors at times $t = t_{\ell_1}, ..., t_{\ell_k}$. At time $t = t_{\ell_k}$, vehicle $i$ carries out the subsequent actions i)-iii). Then go back to Step 2.

i) Set $k$ to $k + 1$ and record the time $t_{\ell_k}$ at which the information is received from neighbors.

ii) Update the velocity information $h_j\left(t_{n_j(\ell_k)}^j\right)$ of the neighbor.

iii) Compute $\tau_{n,k}^i$ by followings. The defination is similar as (11) , and is omitted here. Vehicle $i$ computes

$$\begin{bmatrix} \widetilde{\tau}_{xn,k}^i \\ \widetilde{\tau}_{yn,k}^i \end{bmatrix} = \phi(a_{n,k}^i, b_{n,k}^i, c_{n,k}^i),$$

where

$$\begin{aligned} a_{n,k}^i &:= d_i h_i(t_n^i) - \sum_{j \in \mathcal{N}_i} h_j\left(t_{n_j(\ell_k)}^j\right) \\ b_{n,k}^i &:= \delta_i G(t_{\ell_k}) [1, 1]^T \\ c_{n,k}^i &:= c_{n,k-1}^i + \left(t_{\ell_k} - t_{\ell_{k-1}}\right) a_{n,k-1}^i, \end{aligned} \tag{16}$$

and we can obtain new inter-event time,

$$\tau_{n,k}^i := \min\left\{\min\{\widetilde{\tau}_{xn,k}^i, \widetilde{\tau}_{yn,k}^i\} + (t_{\ell_k} - t_{\ell_0}), \tau_{\max}^i\right\}. \tag{17}$$

Then $t_n^i + \tau_{n,k}^i$ is the candidate of the next triggering time of $i$.

Step 3-b. If vehicle $i$ does not receive any information in the interval $(t_{\ell_k}, t_n^i + \tau_{n,k}^i)$, then it sets $t_{n+1}^i = t_n^i + \tau_{n,k}^i$.

Step 4. Vehicle $i$ sets $n$ to $n + 1$. Then go back to Step 1.

## B. Self-triggered distributed guidance law

In this subsection, we devise a self-triggered, distributed guidance strategy, grounded on relative positions acquired through sensors. To attain the desired formation configuration, we define a formation control error specific to the $i$th vehicle, as elaborated below:

$$z_i = J_i^T(\psi_i)[\sum_{j=1}^M a_{ij}(\eta_{ij} - \eta_{ijd}) + a_{i0}(\eta_{i0} - \eta_{id})], \quad (18)$$

where $\eta_{ij} = \eta_i - \eta_j$, $\eta_{ijd} = \eta_{id} - \eta_{jd}$, the rotation matrix $J_i(\psi_i)$ is expressed as

$$J_i(\psi_i) = \begin{bmatrix} \cos\psi_i & -\sin\psi_i \\ \sin\psi_i & \cos\psi_i \end{bmatrix}.$$

The self-triggered information of the $i$th USV is defined by the following aperiodic scheme

$$[\eta_{ij}^e(t), \eta_{i0}^e(t)] = [\eta_{ij}(t_n^i), \eta_{i0}(t_n^i)], t \in [t_n^i, t_{n+1}^i),$$

where $t_n^i$ is the triggering time with $n \in \mathbb{N}$, $t_{n+1}^i$ is determined by the scheme in III-A 3).

According to (18), we define the self-triggered formation control error as

$$z_i^e = J_i^T(\psi_i)[\sum_{j=1}^M a_{ij}(\eta_{ij}^e - \eta_{ijd}) + a_{i0}(\eta_{i0}^e - \eta_{id})], \quad (19)$$

where the relative state $\eta_{ij}^e$, $\eta_{i0}^e$ can be sampled by the sensors at triggering time.

To mitigate the challenges posed by under-actuation, we introduce an error transformation as follows

$$\bar{z}_i^e = z_i^e + [\Delta_0, 0]^T \in \mathbb{R}^2, \quad (20)$$

where $\Delta_0 \in R$ is a positive constant. Taking the time derivative of (19) yields

$$\dot{\bar{z}}_i^e = C_i[u_i, r_i]^T - r_i B \bar{z}_i^e + \sigma_i, \quad (21)$$

where $C_i = \text{diag}\{d_i, \Delta_0\}$, $B = \begin{bmatrix} 0 & -1 \\ 1 & 0 \end{bmatrix}$, $\sigma_i = -\Sigma_{j \in \mathcal{N}_i} a_{ij} J_i^T(\psi_i)\dot{\eta}_j - \Sigma_{j \in \mathcal{N}_i} a_{ij} J_i^T(\psi_i)\dot{\eta}_{ijd} + [0, d_i v_i]^T + \Sigma_{j \in \mathcal{N}_i} a_{ij} J_i^T(\psi_i)\dot{\tilde{\eta}}_{ijd} - a_{i0} J_i^T(\psi_i)\dot{\eta}_0 - a_{i0} J_i^T(\psi_i)\dot{\tilde{\eta}}_{i0}$, $\tilde{\eta}_{ij} = \eta_{ij} - \eta_{ij}^e$, and $\| \dot{\sigma}_i \| \leq \sigma^*$ with $\sigma^*$ being a positive constant.

Note that $\sigma_i$ is totally unknown due to the unavailable continuous neighboring velocites $\dot{\eta}_j$, $\dot{\eta}_0$ and self-triggered error $\tilde{\eta}_{ij}$. To estimate it, an extended state observer is proposed as

$$\begin{cases} \dot{\hat{\bar{z}}}_i^e = \vartheta_i + \hat{\sigma}_i - p_{i1}\tilde{\bar{z}}_i^e \\ \dot{\hat{\sigma}}_i = -p_{i2}\tilde{\bar{z}}_i^e, \end{cases} \quad (22)$$

where $\tilde{\bar{z}}_i^e = \hat{\bar{z}}_i^e - \bar{z}_i^e$ is the the estimated triggering error, $\vartheta_i = C_i[u_i, r_i]^T$ and $p_{i1}, p_{i2}$ are positive constants. Based on the extended state observer, we formulate a self-triggered, distributed kinematic guidance law as

$$\begin{bmatrix} u_{ir} \\ r_{ir} \end{bmatrix} = C_i^{-1}(\frac{-O_i\bar{z}_i^e}{\sqrt{\|\bar{z}_i\|^2 + \jmath_i^2}} + r_i B \bar{z}_i^e - \hat{\sigma}_i), \quad (23)$$

where $O_i = \text{diag}\{o_{ix}, o_{iy}\} \in \mathbb{R}^{2\times 2}$ with $o_{ix} \in \mathbb{R}$ and $o_{iy} \in \mathbb{R}$ being positive constants; $\jmath_i \in \mathbb{R}$ is a positive constant.

## IV. Main Results

In this section, simulation results are provided to illustrate the self-triggered distributed formation controllers for multiple underactuated USVs with limited communication resources in GPS-denied environments.

Consider a networked system consisting of five USVs. Since there is no thrust in the direction of underactuated ship roll, a small transverse speed will be generated under the influence of wind and wave currents, resulting in sideslip angle. To compensate for this, $v$ is set to 0.2 for all USVs.

The initial states of five USVs are given as $\eta_1 = [4.8, 5.8]^T$, $\eta_2 = [-1.1, 6]^T$, $\eta_3 = [5, 0]^T$, $\eta_4 = [-4, 6]^T$, $\eta_5 = [7, -6]^T$. The desired formation pattern are set as $\eta_{1d} = [0, 0]^T$, $\eta_{2d} = [-5, 5]^T$, $\eta_{3d} = [5, -5]^T$, $\eta_{4d} = [-10, 10]^T$, and $\eta_{5d} = [10, -10]^T$. The parameters for the self-triggered distributed formation controllers are $K_i = \text{diag}\{2, 2\}$, $\Delta_0 = 0.1$, $\epsilon_i = 0.01$, $\omega = 0.1046$, $\delta_1 = \delta_2 = \delta_3 = 0.01$, $\delta_4 = \delta_5 = 0.03$, $\tau_{\max}^1 = \tau_{\max}^2 = \tau_{\max}^3 = 3$s, $\tau_{\max}^4 = \tau_{\max}^5 = 5$s. To avoid Zeno behaviors, we set $\tau_{\min}^1 = \tau_{\min}^2 = \tau_{\min}^3 = 12.483 \times 10^{-3}$s, $\tau_{\min}^4 = \tau_{\min}^5 = 55.857 \times 10^{-3}$s. Vehicles 4 and 5 have larger thresholds and lower bounds for inter-event times due to their smaller number of neighbors compared to the other USVs.

The communication topology among the fleet is depicted in Fig. 3. The simulation results, as shown in Fig. 4-Fig. 7, confirm that the proposed self-triggered distributed controller successfully manages a time-varying formation involving five USVs. Fig. 5 displays the trajectory of self-triggered coordinated errors, and it is shown that these errors converge to a boundary at steady state. The relative straight-line distances between each USV and its neighbors are illustrated in Fig. 6, where it can be observed that the inter-vehicle distances stabilize within a narrow band around the value 7.07. Additionally, Fig. 7 shows that triggerring occurs frequently in the interval [0, 3] but the frequency of triggering decreases after 3s.

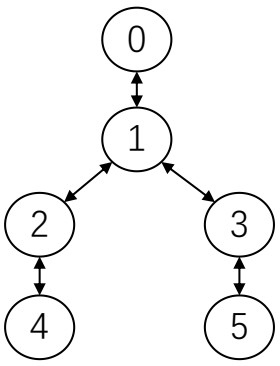

Fig. 3. Communication topology

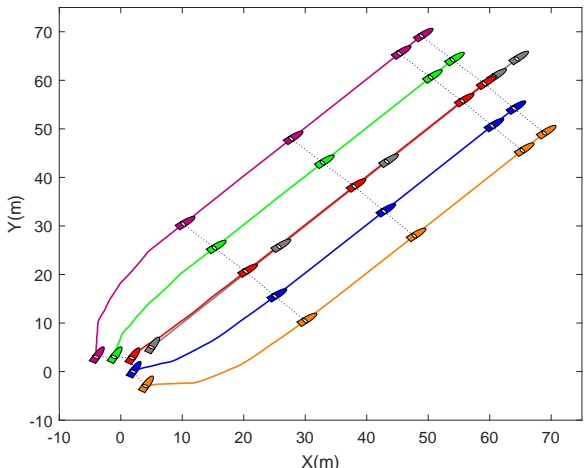

Fig. 4. A formation pattern of five USVs

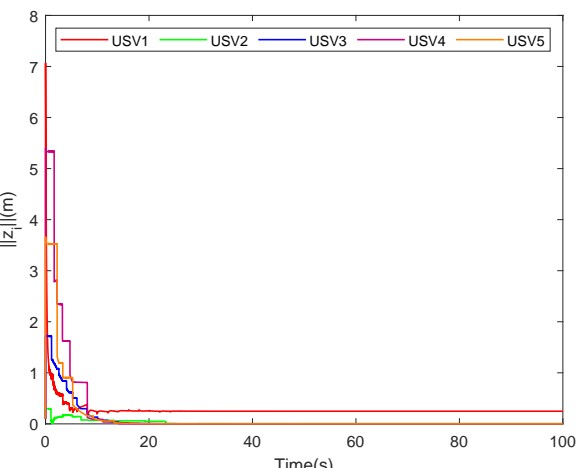

Fig. 5. Formation tracking errors

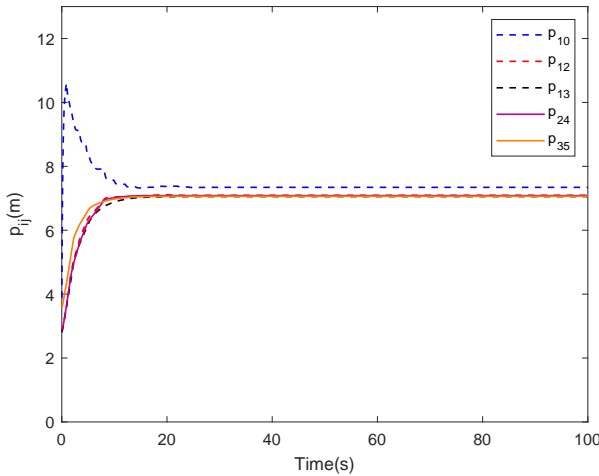

Fig. 6. The relative distances between the USV and its neighbor

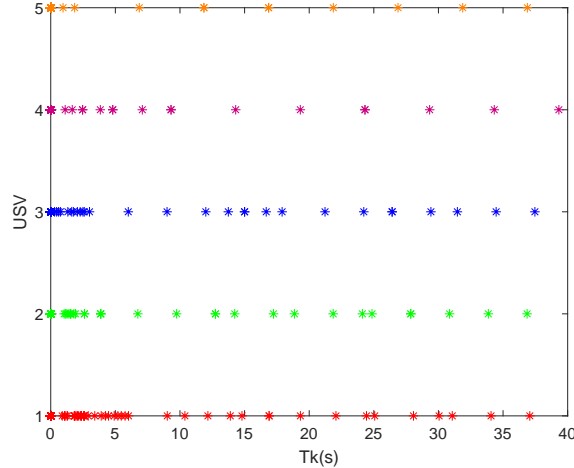

Fig. 7. Triggering times of USVs

## V. Conclusion

In this paper, we consider a self-triggered distributed formation control method for multiple underactuated USVs with limited communication, which uses measurements of relative states and quantized velocity information. A distributed guidance law is developed for each vehicle based on a self-triggered consensus approach and an extended state observer. Compared with previous self-triggered studies, we use relative states instead of absolute ones. Additionally, the estimation of the states of neighbors is not needed, reducing the computational burden on USVs. Therefore, the proposed algorithm can be applied to GPS-denied environments. Meanwhile, since the velocity information is already quantized at the sensing stage, it can be conveyed without any errors even through finite capacity channels. The information transmitted between vessels is the index rather than the actual data, which can prevent information leakage. Simulation results substantiate the effectiveness of the proposed integrated guidance and self-triggered method by relative states for underactuated USVs. In the future, it will be interesting to implement the proposed self-triggered distributed formation control algorithm in a real-world multi-USV system.

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
