# OpenReview forum: "Self-triggered Distributed Formation Control of Under-actuated Unmanned Surface Vehicles in GPS-denied Environments"
_IEEE.org/ICIST/2024/Conference — IEEE ICIST 2024 Conference Submission_

### Official Review · Reviewer_Kwfi · 2024-08-21
**Accept**

**Rating:** 7
**Confidence:** 4

**Review:**

This paper proposes a self-triggered formation control method for situations with missing GPS signals. The velocity quantization-based self-triggered mechanism reduces data exchange frequency through wireless network infrastructure, and the effectiveness of the proposed method is validated through simulation results. However, to further enhance the quality of the paper, the following suggestions should be considered: 1. The self-triggered method emphasized in the main contributions should be compared with similar methods to highlight its effectiveness; 2. Some formulas in the paper, such as Formula 3, exhibit non-standard forms, and some parameters provided lack specified ranges

---

### Official Review · Reviewer_i8f1 · 2024-08-22
**Self-triggered Distributed Formation Control of Under-actuated Unmanned Surface Vehicles in GPS-denied Environments**

**Rating:** 7
**Confidence:** 5

**Review:**

This paper presents a self-triggered distributed formation control method for multiple unmanned surface vehicles (USVs) with limited communication resources in GPS-denied environments. This work is well organized. Below are some comments.
(1)	The contributions should be illustrated in a clearer manner. For example, what is the main improvement of the paper compared to the existing results. The authors should explain the unique contributions of this paper.
(2)	It is better to give a guideline of selection for all control parameters.
(3)	The simulation results should be explained more carefully.
(4)    The self-triggered method  should be compared with other methods to highlight its advantage.
(5)	The paper is well presented, spelled correctly. I recommend authors to carefully read the entire paper to find possible misspellings.

---

### Official Review · Reviewer_pPzE · 2024-08-22
**Accepted, but some recommendations remain**

**Rating:** 7
**Confidence:** 5

**Review:**

This paper designs a self-triggered distributed formation control method for multiple unmanned surface vehicles. The self-triggering mechanisms proposed in this paper are effected by comparing the relative states between vehicles, which reduces the computational burden and eliminates the need for absolute states. Meanwhile, secure transmission tasks achieved by the fact that the information transmitted between vessels is not actual data. There are still some  suggestions:
1. The comparative simulation of the event triggering methods presented in [16], [17] could be added.
2. This paper mentions that this is the first attempt to consider the self-triggering approach in the design of distributed formation controllers for USVs. Hopefully, the author will conduct more research to determine if this fact holds true.

---

### Official Review · Reviewer_XskZ · 2024-08-24
**The article proposes a self triggering communication method that uses relative position information to predict the next triggering time for unmanned ship formation control in environments with weak GPS signals. The selected topic is in line with the research background in the field of control and has good research significance. In order to better publish this paper, the following suggestions are proposed:**

**Rating:** 8
**Confidence:** 4

**Review:**

1) There are multiple missing spaces and incorrect use of singular and plural words in the article.
2) There are some symbol font issues in the article, where the N-symbol font representing the set is inconsistent before and after.

---

### Decision · Program_Chairs · 2024-09-06

Accept (Oral)